# Identification of Missense *ADGRV1* Mutation as a Candidate Genetic Cause of Familial Febrile Seizure 4

**DOI:** 10.3390/children7090144

**Published:** 2020-09-18

**Authors:** Ji Yoon Han, Hyun Joo Lee, Young-Mock Lee, Joonhong Park

**Affiliations:** 1Department of Pediatrics, College of Medicine, The Catholic University of Korea, Seoul 06591, Korea; han024@catholic.ac.kr; 2Departments of Pediatrics, Yonsei University College of Medicine, Seoul 06273, Korea; GENEJOO@yuhs.ac; 3Department of Laboratory Medicine, College of Medicine, The Catholic University of Korea, Seoul 06591, Korea; 4Department of Laboratory Medicine, Jeonbuk National University Medical School and Hospital, Jeonju 54907, Korea

**Keywords:** *ADGRV1* mutation, familial febrile seizure 4, afebrile seizure, febrile seizure, targeted exome sequencing

## Abstract

Febrile seizure (FS) is related to a febrile illness (temperature > 38 °C) not caused by an infection of central nervous system, without neurologic deficits in children aged 6–60 months. The family study implied a polygenic model in the families of proband(s) with single FS, however in families with repeated FS, inheritance was matched to autosomal dominance with reduced disease penetrance. A 20 month-old girl showed recurrent FS and afebrile seizures without developmental delay or intellectual disability. The seizures disappeared after 60 months without anti-seizure medication. The 35 year-old proband’s mother also experienced five episodes of simple FS and two episodes of unprovoked seizures before 5 years old. Targeted exome sequencing was conducted along with epilepsy/seizure-associated gene-filtering to identify the candidate causative mutation. As a result, a heterozygous c.2039A>G of the *ADGRV1* gene leading to a codon change of aspartic acid to glycine at the position 680 (rs547076322) was identified. This protein’s glycine residue is highly conserved, and its allele frequency is 0.00002827 in the gnomAD population database. *ADGRV1* mutation may have an influential role in the occurrence of genetic epilepsies, especially those with febrile and afebrile seizures. Further investigation of *ADGRV1* mutations is needed to prove that it is a significant susceptible gene for febrile and/or afebrile seizures in early childhood.

## 1. Introduction

The international League Against Epilepsy defined febrile seizures (FSs) as a seizure occurring in children aged 6 to 60 months related to a febrile illness (temperature > 38 °C) not caused by an infection of the central nervous system (CNS), without previous afebrile seizures or neurologic deficits [1,2]. The mean prevalence of FS in children aged 6–60 months according to hospital visit rates in South Korea was 6.92% (6.12% for girls; 7.67% for boys) during the period 2009–2013 [3]. About a third of patients with experience of FS will have a second attack, and 50% of those will have a third episode. Past history of FS during childhood has been reported in 5 to 15% of epileptic patients, with a higher risk of evolving epilepsy found in children < 14 years with preceding complex FS [4]. The risk of unprovoked seizures is 5 to 7 times higher in children who experience FS compared to the general population and is estimated to be 2 to 5% [5,6]. Prolonged FS and febrile status epilepticus are considered as potential risk factors in subsequently developing epilepsy [7]. Simple FS seem to be genetically complex disorders affected by variations in several susceptibility genes [8]. Children with a familial history of FS show a three-fold or higher risk of recurrence than the general population experiencing FSs. A positive familial history within first-degree relatives is regarded as a compatible risk factor for recurrent FSs than recent infections, fever, and perinatal exposure [9,10]. The family study implied a polygenic model in families of proband(s) with a single FS, however, in the families with repeated FSs, the inheritance was matched to autosomal dominance with reduced disease penetrance [11].

To date, the genetic heterogeneity of familial FSs (termed FEB1–FEB11) has been described in different studies [12,13,14,15,16,17,18,19,20,21,22,23,24,25]. Particularly, *ADGRV1* (OMIM *602851) encoding adhesion G protein–coupled receptor (aGPCR) V1, a large calcium-binding protein widely expressed in the CNS, was previously reported as a genetic cause of afebrile and febrile seizures [26]. In this study, targeted exome sequencing was conducted along with epilepsy/seizure-associated gene-filtering to identify the candidate causative mutation in a Korean family with febrile and afebrile seizures.

## 2. Case Presentations

A 20 month-old girl (II-2 in Figure 1a) was referred to the department of pediatric neurology after four episodes of FS. Her mother witnessed generalized tonic–clonic seizures that lasted under 5 min with a fever of 38 °C–40 °C. By the time the patient was transported to the emergency department, she had fully recovered. At 21 months of age, she came to our emergency department after generalized tonic–clonic seizure that lasted for 20 min without fever. The seizure had ceased and her mental state had recovered when she arrived at the emergency room. Two hours later, her seizure recurred and the seizure was terminated after an injection of lorazepam. In brain magnetic resonance images (MRI), no obvious abnormalities for her age including an appropriate myelination pattern were observed (Figure 1b). The interictal electroencephalogram (EEG) was normal (Figure 1c). The results of metabolic laboratory testing including plasma amino acid, thyroid function tests, lactate/pyruvate, urine organic acid, and blood gases were normal. We observed her without anti-seizure medication at the outpatient clinic. After that, she showed two more simple FSs at the age of 24 months and 35 months. At 36 months old, she was admitted because of a prolonged FS. The seizure apparently lasted for 25 min and ceased after the administration of intravenous lorazepam in the emergency department. Later, a fever of 39.5 °C was noted, and adenoviral infection was diagnosed. Her growth and developmental milestones were proper for her age. Her height (92 cm), weight (13 kg), and head circumstance (47 cm) were 50 percentiles, respectively. Febrile and afebrile seizures had not occurred after the age of 3 and her development was appropriate for her age. Except for a history of seizures in her mother (I-2 in Figure 1a), there was no family history of developmental delay, intellectual disability, and epilepsy. The 35 year-old proband’s mother experienced five episodes of simple FS and two episodes of unprovoked seizures before 5 years old. The mother’s computed tomography scan and EEG were normal and she has remained seizure free since 5 years old without anti-seizure medication. The mother’s development was completely normal and she completed her college education. 

## 3. Molecular Analysis

### 3.1. Targeted Exome Sequencing

The study protocol was approved by the Institutional Review Board of the Catholic University of Korea. Written informed consent was collected from the parents on behalf of their children for the publication of recognizable data or images included in this report before the blood sampling, and clinical data were obtained from the proband and her parents. To resolve the potential genetic cause, the genomic DNA of the proband was studied by targeted exome sequencing using TruSight One Sequencing Panel (Illumina, Inc., San Diego, CA, USA), which covers rare inherited disease-associated regions of the exome with a comprehensive coverage of > 4800 disease-associated genes. Massively parallel sequencing was conducted using the Illumina HiSeq2500 (Illumina, Inc.) to generate paired-end reads of 150 nucleotides. Exome sequences were estimated for all modes of inheritance and the variants filtered initially for rare variants (allele frequency < 0.01) from the public sequence databases (gnomAD, https://gnomad.broadinstitute.org). The remaining base change and small indels located on epilepsy/seizure-associated genes were selected according to web-based genetic databases (OMIM, https://www.omim.org/; ClinVar, https://www.ncbi.nlm.nih.gov/clinvar/). The candidate missense mutations were estimated to be pathogenic or have a damaging effect by in silico analysis for conservation level as well as for functional effect.

### 3.2. mRNA Expression Analysis for ADGRV1 Mutation

To determine the pathogenicity of the candidate *ADGRV1* mutation, quantitative reverse transcription polymerase chain reaction (RT-PCR) was performed with complementary DNA synthesized from RNA isolated from whole blood on a CFX96™ Real Time PCR Detection System (BioRad, Hercules, CA, USA) using the 5′ reporter FAM™ dye-labeled and 3′ quencher Minor groove binder (MGB)-labeled probes (Applied Biosystems, Foster City, CA, USA). Gene expression level was normalized to *GAPDH* endogenous control and analyzed according to the relative quantification method (2^−ΔΔCt^). Three independent experiments were performed. Mean difference in gene expression between p.Asp680Gly and the wild-type of the *ADGRV1* were estimated by Student’s *t*-test. The *p* value < 0.05 was considered to indicate a statistically significant difference.

## 4. Results

By estimating the sequence quality along all sequences, 18.9 million reads 150 bp in read length were produced from the patient sample. The % bases above average 30× were achieved for 99.2% of the target region and the mean read depth (×) was 170 bp. Targeted exome sequencing identified a heterozygous c.2039A>G of the *ADGRV1* gene leading to a codon change of aspartic acid to glycine at position 680 (NM_032119.3: c.2039A>G, p.Asp680Gly; rs547076322) had not been reported previously to be related to afebrile and/or febrile seizures in the proband. To confirm the mutation segregated with affected members, Sanger sequencing was conducted and revealed that this mutation as a heterozygous state was present in the proband and her mother, respectively (Figure 2a). Cross-species sequence comparisons (phastCons, SiPhy, and GERP) of amino acid sequences of ADGRV1 protein revealed that this mutated site was highly conserved in vertebrates (phastCons 0.994 > cut-off of 0.8, SiPhy 16.285 > 12.17, and GERP 5.87 > 4.4) [27]. The exome database of gnomAD showed a rare allele frequency of 0.00002827. In addition, quantitative RT-PCR revealed that the expression of *ADGRV1* mutant was weaker than that of the wild type (*p =* 0.048) (Figure 2b). Particularly, (likely) pathogenic *SCN1A* mutations were not identified in this family.

## 5. Discussion

Seizures in FEB 4 are related to febrile episodes in childhood with no evidence of defined pathologiccause or CNS infection. Simple FS is common, affecting 2–5% of children aged from 6 to 60 months. The tendency of developing epilepsy following simple FSs is not common. Complex FS is characterized as focal onset, with a duration of >15 min, and/or >one seizure in 24 h and related to an increased incidence of epilepsy. Channelopathies caused by mutation in ligand-gated or voltage-gated channels lead to idiopathic epilepsies such as FS, benign neonatal or infantile epilepsy, and autosomal dominant nocturnal frontal-lobe epilepsy. FEB 4 locus comprise the *ADGRV1* gene, the causative gene for audiogenic reflex seizures in the Frings mouse (*Adgrv1*) [28]. In this study, *Avlgr1* (*Algr1b, Algr1d*, and *AVlgr1e*) mRNA appeared dorminantly in the neuroepithelium of the developing mouse brain. Knockout mice without exons 2–4 of *Avlgr1b* were prone to experience audiogenic reflex seizure, although obvious cerebral histological abnormalities were not proven [29]. Differing from previous patients with ultra-rare *ADGRV1* variants [26,30], our patient showed febrile seizures with afebrile seizures. 

To the best of our knowledge, we reported a first Korean family who showed FS and afebrile seizures with *ADGRV1* mutation, but without developmental delay/and or intellectual disability and the seizure disappeared after 60 months without anti-seizure medication. The identified nucleotide change in *ADGRV1* replaces aspartic acid with glycine at position 680 of the ADGRV1 protein (p.Asp680Gly). This protein’s glycine residue is highly conserved, and has not been reported for individuals with *ADGRV1*-related FEB 4 (Table 1). In a previous study, mutation screening for the *ADGRV1* detected several missense mutations with allele frequency < 0.0005 in 48 families with FSs or afebrile seizures [26]. Although different missense mutations (p.Tyr674Cys and p.Val754Ala) at the near codons have been predicted to be benign or pathogenic, the segregation of the mutation in our family supports its pathogenicity as in autosomal dominant manner. However, this prediction has not been confirmed by functional analysis, and the clinical manifestation of the observed mutation is certain only in two-generation family members showing afebrile and febrile seizures. Thus, this *ADGRV1* mutation may be a candidate as a genetic cause of afebrile and febrile seizures in FEB 4.

On the other hand, *ADGRV*1 variation plays a part in developing epilepsy with myoclonic seizures, although the inheritance manner may be different in various patients [30]. The function of *ADGRV1* is unestablished, but multiple calcium exchanger b-repeats in the ectodomain imply a contribution to protein–protein interaction that may be calcium mediated [31]. Particularly, the sequence of the *ADGRV1/VLGR1* gene harbors the same EAR (epilepsy-related repeat) domain located in the *LGI1* (leucine-rich glioma-inactivated 1, 604619) gene which is a causative gene in autosomal dominant lateral temporal lobe epilepsy with auditory features (ADLTE; 600512) [32]. *ADGRV1* and *LGI1* encode proteins that share a seven-fold repeated 44-residue motif homology domain, which has been named an EAR domain (Figure 3a) [32]. Scheel et al. suggested that the EAR domain plays a critical part in developing epilepsy by attaching to an unknown antiepileptic ligand or by interfering, with synaptogenesis or axon guidance [32]. The pathogenesis of *ADGRV1* haploinsufficiency leading to seizures remains not well established, but animal studies have provided early insight into the nature of the mechanism. Comprehending the signaling pathways downstream of aGPCRs is not only essential for drug discovery, but also for achieving a fundamental understanding of receptor function [33]. ADGRV1, the largest aGPCR, has been found to couple to Gαi and signal to protein kinases A and C via Gαq and Gαs (Figure 3b). Lobe-Philippot et al. reported that *ADGRV1* is needed for GABAergic interneuron development in the auditory cortex [34]. Dysfunction of cortical GABAnergic neurons can be a potential epileptogenic mechanism in humans, thus it can be a therapeutic target. In this family, the proband and her mother showed recurrent febrile seizures without other neurological comorbidity, and the seizure disappeared after the age of 5 without anti-seizure medication. *ADGRV1* can be a susceptible gene for the mutation that can lead to febrile and afebrile seizures. On the other hand, whole-exome sequencing in combination with a genome-wide association study might be the most straightforward way to get some insight into the genetics of FS, which addresses the issues of phenocopies and genetic heterogeneity by sheer statistical power and sample size [35,36].

## 6. Conclusions

*ADGRV1* may be related to a range of self-limited, mild febrile or infantile seizure without intellectual disability and developmental delay. *ADGRV1* mutation may play an influential role in the occurrence of genetic epilepsies, especially those with febrile and afebrile seizures. If children under the age of 5 with no other neurological symptoms have febrile and other types of seizures, including those with no fevers, the ADGRV1 test will be useful like other genetic testings in finding the causes and prognosis of seizures. This may provide reassurance for the family towards an individual patient based on their genetic profile instead of empirical trials of anti-seizure medication. Further research is needed to establish an association between *ADGRV1* mutations and febrile/afebrile seizures. 

## Figures and Tables

**Figure 1 children-07-00144-f001:**
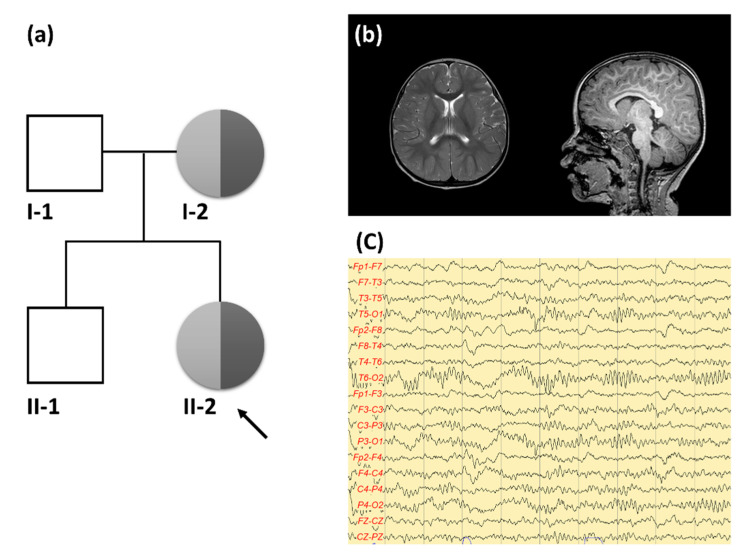
(**a**) Pedigree of a Korean family with afebrile (light grey) and febrile seizures (dark grey) carrying *ADGRV1* mutation. (**b**) Brain magnetic resonance imaging was done at the age of 2 in the proband. No abnormal signal intensity or enhanced lesion in the brain parenchyma was seen. No evidence of abnormal finding in the diploic space and cerebrospinal fluid space. (**c**) Interictal electroencephalogram was normal tracing at the age of 2 in the proband. Focal abnormalities, persistent asymmetries, and potentially epileptogenic discharges were not seen.

**Figure 2 children-07-00144-f002:**
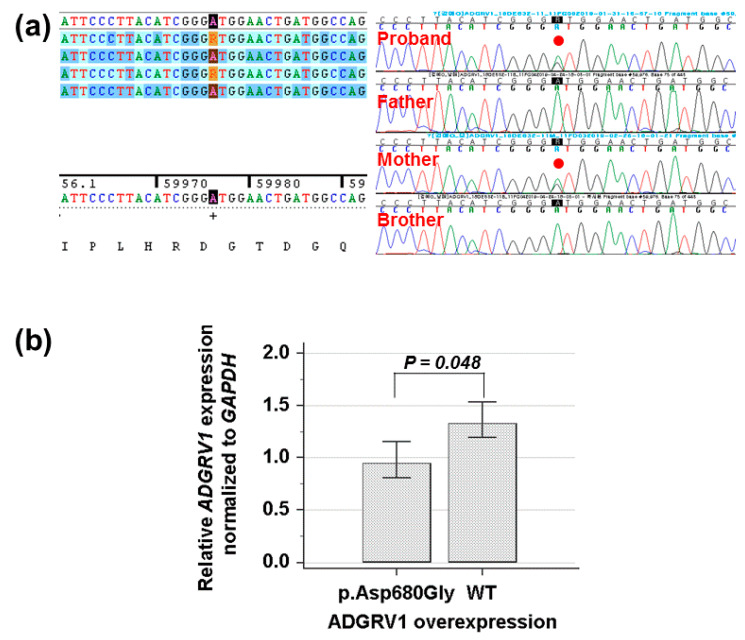
(**a**) Sanger sequencing shows a heterozygous missense mutation (NM_032119.3: c.2039A>G, p.Asp680Gly) of the *ADGRV1* gene in the proband and her family member, respectively. (**b**) Quantitative reverse transcription polymerase chain reaction indicates that the missense mutation (p.Asp680Gly) decreases the mRNA expression level of the *ADGRV1*, compared to the wild type (*p =* 0.048).

**Figure 3 children-07-00144-f003:**
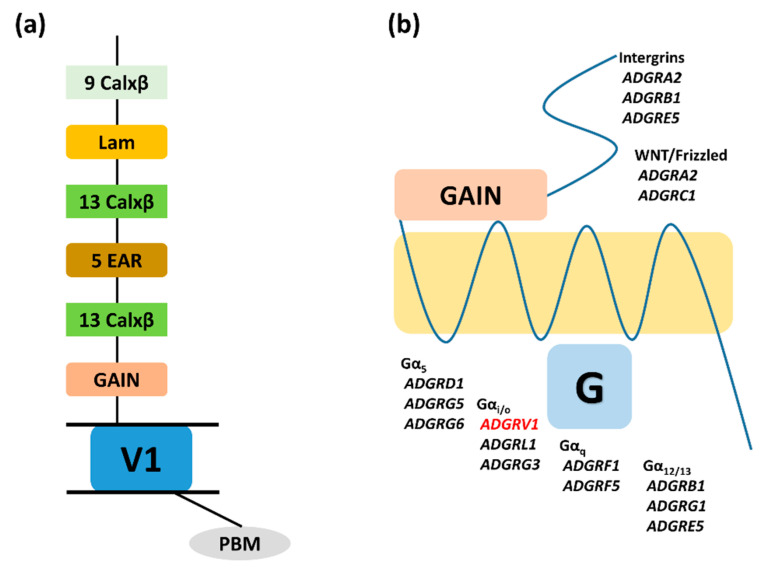
(**a**) Adhesion G protein-coupled receptor structure (GPCR). Calx, calcium exchanger; Lam, laminin; GAIN, GPCR autoproteolysis inducing domain; PBM, PDZ binding motif (**b**) Adhesion G protein-coupled receptor signaling pathways. GAIN, GPCR autoproteolysis–inducing domain; GPCR, G protein–coupled receptor.

**Table 1 children-07-00144-t001:** Results of in silico analysis of rare missense *ADGRV1* mutations with allele frequency < 0.0005 reported by Myers KA et al. and this study.

Base Change	Codon Change	rs ID	gnomAD	Polyphen-2	Mutation Taster	Grantham (<100)	GERP (>4.4)	Protein Domain	Inheritance
c.2021A>G	p.Tyr674Cys	rs761220696	4/17562 (EA) 1/110184 (EU) 5/242476 (All)	Benign (0.001)	Polymorphism	Radical 194	−5.21	Calx-β5	Father
* c.2039A>G	p.Asp680Gly	rs547076322	7/17866 (EA) 7/247616 (All)	Damaging (1)	Disease causing	Conservative 94	5.87	Calx-β5	Mother
c.2261T>C	p.Val754Ala	rs374609813	0/19516 (EA) 22/30544 (SA) 13/128162 (EU) 39/280148 (All)	Damaging (0.682)	Disease causing	Conservative 64	5.09	Extracellular, links Calx-β5 and Calx-β6	Unknown
c.5722G>A	p.Asp1908Asn	rs757418364	0/19490 (EA) 9/128042 (EU) 9/279986 (All)	Damaging (0.976)	Disease causing	Conservative 23	5.56	Calx-β13	Mother
c.8266G>A	p.Gly2756Arg	rs546198768	0/18314 (EA) 11/117330 (EU) 15/257862 (All)	Damaging (1)	Disease causing	Radical 125	5.15	Calx-β19	Mother
c.9466A>G	p.Ile3156Val	rs372484022	0/19532 (EA) 57/30568 (SA) 7/127064 (EU) 69/279132 All)	Benign (0.274)	Disease causing	Conservative 29	1.86	Calx-β22	Mother
c.13228G>A	p.Glu4410Lys	rs371970388	0/19532 (EA) 17/126258 (EU) 18/273998 (All)	Damaging (0.676)	Polymorphism	Conservative 56	1.39	Calx-β30	Unknown
c.13495C>T	p.Arg4499Cys	rs567519802	0/17780 (EA) 6/29982 (SA) 7/245774 (All)	Damaging (0.998)	Disease causing	Radical 180	5.97	Extracellular, links Calx-β30 and Calx-β31	Unknown
c.13568G>C	p.Ser4523Thr	rs376673439	0/17816 (EA) 11/111494 (EU) 12/246276 (All)	Damaging (0.996)	Disease causing	Conservative 58	5.78	Calx-β31	Father

* Identified in the present study. Reference sequence number: NM_032119.3 NP_115495.3. Abbreviations: rsID, reference ID in dbSNP; gnomAD, The Genome Aggregation Database; EA, East Asian; EU, non-Finnish Europe; SA, South Asian; All, total population; Polyphen-2, Polymorphism Phenotyping v2;) AF (Allele Frequency); Grantham, Grantham scores for conservative < 100; GERP, Genomic Evolutionary Rate Profiling for conservative > 4.4.

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
