# Peer review of "Identification of Missense ADGRV1 Mutation as a Candidate Genetic Cause of Familial Febrile Seizure 4"

_children, 2020, doi:10.3390/children7090144_

Round 1

Reviewer 1 Report

Remarks

1.The sentence concerning EEG is not clear:

The EEG between seizures was with in normal

  1. How the authors explain the observation of the child after afebrile seizures without treatment? I can not find any information about afebrile seizures (time of appearance anf morphology). How many times the EEG was performed? What were the EEG results especially after afebrile seizures?
  2. The sentences are not clear ( especially: time of appearance of simple FS, duration of complex FS):

Simple FS is common, affecting 2%-5% of children from 3 months to 5 years. Complex FS is characterized as focal onset, with duration of > 30 minutes, and/or > one seizure in 24 hour and related to an increased incidence of epilepsy.

  1. The genetic panel covered a 12 Mb region spanning 4,813 genes with clinical relevance: what does it exactly mean? Mutations in SCN1 genes were excluded?

Author Response

Manuscript ID: children-895259
Title: Identification of missense ADGRV1 mutation as a candidate genetic cause of familial febrile seizure 4

Dear Editor and Reviewers of CHILDREN

Thank you for your considerations of our manuscript entitled “Identification of missense ADGRV1 mutation as a candidate genetic cause of familial febrile seizure 4” and for giving us an opportunity to revise the manuscript. The changes to our manuscript within the document were highlighted by using red colored text. We expect our article to be appropriate to the publication in CHILDREN. The electronic word file containing respond to decision letter was attached as a cover letter.

Once again, thank you for a consideration about our manuscript to CHILDREN and we look forward to receiving your reply.

Best regards,

Ji Yoon Han and Joonhong Park

Reviewer 2 Report

Thank you for submitting the manuscript “Identification of missense ADGRV1 mutation as a candidate genetic cause of familial febrile seizure 4” and highlighting an exciting finding. In reviewing the manuscript, I thought that it was overall well written, but some areas need clarification or elaboration. 

  1. The title is good, and the abstract is succinct.
  2. Current literature demonstrates that there may be a strong genetic component, and there is increasing evidence for susceptibility genes in febrile seizures. The statement “Further investigation of ADGRV1 mutations suggests that it will prove to be a significant susceptible gene for febrile seizure and/or afebrile seizure in early childhood.” is a more explicit statement. I hope that the author would consider altering the statement and be amenable to a less definite statement as this is a single two-generation family study?
  3. Family studies may be an exciting entry to research in the role of ADGRV1 gene mutation, but not sure if ADGRV1 mutation will withstand the test of time and replication. The author should highlight this point and the need for further progress in the genetics of febrile seizures, which may be addressed using an association study, or the author should highlight other ways to approach.
  4. The author has nicely summarized the genotype of ADGRV1 related FEB4 to highlights their essential point. They should also summarize the clinical findings of ADGRV1 related FEB 4 phenotypes and contrast them with this case.
  5. Can the author elaborate further on why they feel this specific mutation (FEB 4 locus) will prove to be a crucial susceptible gene in Febrile seizure, unlike other febrile seizure loci (FEB1-11)?
  6. The references are appropriate, and the figures are adequate. 
  7. The conclusions must not only contain a general interpretation of the findings as well as provide about possible future implications of this study findings.

Author Response

(The authors gave the same response as above.)

Round 2

Reviewer 1 Report

I have just one question regarding your manuscript. The SCN1A mutation was not detected because it was not checked or was checked and excluded?

Author Response

I have just one question regarding your manuscript. The SCN1A mutation was not detected because it was not checked or was checked and excluded?

Reply: The SCN1A gene was included in TruSight One Sequencing Panel used in this study. However, pathogenic mutation of the SCN1A was not identified, but benign variants were detected only as below: 

(Reference sequence: NM_006920.4; Read depth > x30)

Base change       dbSNP147   Genotype
c.3166G>A           rs2298771    homozygous
c.2914-41C>T      rs7601520    homozygous
c.2913+56A>G     rs2020318    homozygous
c.2383-37A>C      rs2126152    homozygous
c.2259T>C           rs6432860    homozygous
c.2143+44C>T     rs75022359   heterozygous
c.1663-47T>G      rs6753355    homozygous
c.1377+92T>G     rs538921      homozygous
c.1377+52G>A     rs6432861    heterozygous
c.1212A>G           rs7580482    homozygous
c.1029-68C>T      rs1461193    homozygous
c.1028+21T>C     rs1542484    heterozygous
c.965-21C>T        rs994399      homozygous
c.603-91G>A        rs3812718    heterozygous
c.383+66T>C        rs8191987    heterozygous
c.-49-34A>T         rs566839       homozygous

Reviewer 2 Report

The manuscript has improved and revision now is acceptable.

Author Response

The manuscript has improved and revision now is acceptable.

Reply: Dear Reviewer of CHILDREN

Thank you for your hard work.

Sincerely, 

J Park